# Revisiting Oral Antiseptics, Microorganism Targets and Effectiveness

**DOI:** 10.3390/jpm13091332

**Published:** 2023-08-29

**Authors:** Lisetty Garrido, Patrícia Lyra, Joana Rodrigues, João Viana, José João Mendes, Helena Barroso

**Affiliations:** Egas Moniz Center for Interdisciplinary Research (CiiEM), Egas Moniz School of Health & Science, Caparica, 2829-511 Almada, Portugalplyra@egasmoniz.edu.pt (P.L.); jpm.viana.1@gmail.com (J.V.); jmendes@egasmoniz.edu.pt (J.J.M.)

**Keywords:** antiseptics, oral microbiota, oral biofilms, dental practice

## Abstract

A good oral health status is mostly dependent on good oral hygiene habits, which knowingly impacts systemic health. Although controversial, chemical oral antiseptics can be useful in adjunct use to mechanical dental plaque control techniques in the prevention and management of local and overall health and well-being. This review aims to revisit, gather and update evidence-based clinical indications for the use of the most popular oral antiseptics, considering different types, microorganism targets and effectiveness in order to establish updated clinical recommendations.

## 1. Introduction

The oral cavity is a complex system with a vast and complex biodiversity, containing up to 800 microbial species [1]. This commensal oral microbiota is present throughout life and can be dysbioticTally perturbed, with certain species overgrowing and increasing the risk of oral disease, particularly dental caries and/or periodontal disease [2,3,4]. Both conditions are considered to be the most prevalent non-communicable diseases worldwide [5]. Therefore, mechanical control of dental biofilm through routine oral hygiene is key to pathogen control and disease prevention [6,7,8,9]. Chemical plaque control with antibacterial mouthwashes can be an alternative to mechanical methods when they are ineffective [7,10,11,12,13,14]. Hence, the use of chemical and mechanical control is recommended synchronously to guarantee good oral hygiene and potentiate a good oral health state [7,15,16,17,18,19,20,21,22].

There is a wide range of commercially available mouthwashes with effective antimicrobial properties for the treatment of oral diseases [17,22,23]. These chemicals inhibit bacterial proliferation or prevent bacterial adherence to tooth surfaces [24] and remove some of the plaque during rinsing, preventing biofilm formation [17,25]. Chlorhexidine (CHX), triclosan (TRC), cetylpyridinium chloride (CPC), and some essential oils (EOs) (eucalyptol, thymol, menthol, etc.) are among the active ingredients commonly found in mouthwash formulations [12].

Furthermore, the condition of the oral cavity is closely linked to general health and well-being [26,27]. Accordingly, oral disinfectants may also play a preventive role in the development and/or spread of some infectious diseases, which can be caused by viruses such as herpes simplex virus (HSV) [28,29], human immunodeficiency virus (HIV), hepatitis B virus (HBV) [28] and more recently COVID-19 [30,31]. Certain mouthwashes are useful against fungal infections such as candidiasis [32,33,34] while also providing an alternative therapy for the treatment of halitosis [35,36].

In addition, oral antiseptics also appear to be a viable option when used as an adjunct to antibiotics for both prevention and treatment of bacterial infections such as ventilator-associated pneumonia (VAP) [37,38], as the oropharyngeal microflora plays a role in the development of nosocomial infections such as VAP [39,40,41]. This form of pneumonia is the most common nosocomial infection in intensive care, affecting 10–30% of mechanically ventilated patients [42,43]. Moreover, both dental professionals and patients are at increased risk of cross-infection in clinical practice [12,17,20,22,44]. The main routes for the spread of most viral, bacterial and fungal infections in a dental setting are droplets, direct contact and air [45,46]. In addition to general biosafety measures, antimicrobial mouthwashes can be used to minimize the oral bacterial burden and the risk of cross-infection [47,48].

This study revisits evidenced-based clinical indications of the use of oral antiseptics, considering different types, microorganism targets and effectiveness.

## 2. Major Oral Antiseptics in Current Clinical Practice

With the aim of managing oral biofilms, a variety of oral hygiene products have been formulated and commercialized [22,23]. Chemical agents in mouthwashes must be efficient at changing the microbiota through selective elimination of pathogens while not adversely disrupting the normal flora [49]. Oral antiseptics contain a mixture of different substances in a solution [50]. Among the different families of oral antiseptics, the following stand out: cationic surfactants derived from bisbiguanides, bispyridinamines and quaternary ammonium derivatives, iodine compounds, phenolic agents, alcohols, and various mixtures of organic compounds [10,12,50,51,52,53,54]. The most representative of these oral antiseptics, their mechanism of action and main adverse effects, among other characteristics, are shown in Table 1.

PVP-I is considered an effective first-line option for the treatment and even prevention of skin infections, oral complications resulting from cancer treatment, common upper respiratory tract infections and regular dental conditions [56]. It is the most frequently used preoperative skin antiseptic in clinical practice settings and has a large spectrum of microbial activity and low potential for resistance [57]. Iodine also protects the cell from the damage caused by free radical oxygen species, contributing to anti-inflammatory properties [56]. When scaling and root planning, the adjuvant use of PVP-I may increment the clinical pocket depth reduction, even though the clinical implication is small to moderate [58]. Most frequently, PVP-I is used for gargling and rinsing with 10–15 mL undiluted for a minimum of 30 s for treatment and prevention of sore throats and prophylactic use before, during and after surgery, as well as part of routine hygiene practices [56].

CHX is the oral antiseptic of choice [10,93]. This has a high substantivity and enormous adherence to the epithelium—characteristics that are not found with this quality in other compounds shown in the table [94]. However, CHX, despite having enormous advantages and therefore being named the gold standard, is not devoid of undesirable effects [10,94]. Adverse effects may be mediated by a wide range of external factors, with smoking standing out as the most common [10,93]. Individuals undergoing treatment with CHX may develop coloration of the skin and mucous membranes [10,28,62]. Nevertheless, new lines of research have been focused on the association between CHX and anti-discoloration systems that provide a decrease in the stains present on tooth surfaces without significantly influencing their efficacy [94,95]

Regarding TRC, it is a nonionic phenolic derivative with known antimicrobial capabilities that has been broadly used as well [53]. Through the mechanisms described in the table, it damages the microorganism’s inner membrane [66]. The highest antimicrobial effects are achieved when TRC is combined with other copolymers, enhancing its substantivity [53,65]. Nevertheless, considerable attention has been given to the potential long-term side effects that this agent can cause, such as increased microbial resistance [53,54,96].

Among the quaternary ammonium compounds, several studies were found on the antiseptic capacity of CPC [12,52,76,97,98,99,100], but this was not the case with BTC and BAC. Although they are known for their antimicrobial capabilities [73,74], there is a lack of rigorous scientific studies on their effects, specifically in the oral cavity. Therefore, the information included in the table on BTC and BAC requires further studies to support it.

Concerning essential oils, the study of their effects has been a challenge due to the obstacle of isolating the active compounds and assuring standardization of components since various factors can differ, influencing chemical composition [81]. They also have manifold psychological effects, such as decreasing anxiety and depression and assisting with falling asleep [81,101]. In dentistry, EOs could be appropriate as preoperative rinses, in periodontal procedures, in post-treatment applications or as a conventional mouthwash [79]

About DEL, it was formulated as an agent with reduced antimicrobial activity and to prevent the disruption of healthy oral microbial flora [84]. This compound inhibits the formation of new plaque by counteracting the enzymes involved in the stability of the plaque matrix [85]. Although tooth staining is reported as an adverse effect of DEL, it is greatly minor when compared to CHX [102]. Currently, DEL has shown value as an adjunctive for treating peri-implant disease [86].

Another substance that has emerged into the spotlight is OCT [51]. It has antimicrobial properties comparable with CHX [51,87,88] but with much less registered adverse effects [51,87]. OCT was recently reported as less cytotoxic than CHX, EOs or PVP-I against gingival fibroblasts and epithelial cells [103]. In the past, OCT was not incorporated into daily routines due to its unpleasant taste, but this was overcome by the addition of flavoring compounds [88].

There has been an increased public interest in herbal health products due to the trend to “go natural” since they are not tested on animals, have less or non-adverse effects, are vegan friendly, contain no added artificial colors or flavors, and for cultural reasons [104,105]. Herbal products were not included in the table since their properties and characteristics are heterogeneous and diverse. Among the most usual herbal ingredients incorporated in oral hygiene products are sanguinarine, propolis, Azadirachta indica (neem), Salvadora persica and Camellia sinensis [11,106]. Numerous herbal or plant extracts possess anti-inflammatory, antipyretic, analgesic, antibacterial, antiviral, anticarcinogenic and antioxidant effects by means of in vitro, in vivo, and animal studies, making its action non-specific [11,106,107]. There is no scientific support to recommend herbal mouthwashes for daily use or for any specific condition but considering the long-term adverse effects of the use of CHX, they are a suitable choice [11]. For example, Salvadora persica extracts and guava mouth rinse can be indicated as phytotherapeutic alternatives to CHX for maintaining gingival health, especially for long-term use [108,109]. These products have become a beneficial option to prevent and treat oral health conditions for rural populations with low socio-economic levels, particularly in low-income countries [11]. Individual herbs proved to have moderate antiseptic effects, and thus, combining various herbs and chemicals exerts a synergic action and increases their antibacterial mechanisms [53,110,111]. Some formulations with herbal ingredients, such as miswak and neem extracts, can be equally effective as dentifrice formulations with chemical antimicrobial agents, like sodium monofluorophosphate and sodium fluoride [53]. When compared with classical chemical antiseptics, plant-derived extracts have better toxicity profiles for humans [106,111]. Although consumers of these products believe they have no adverse effects, there are reports of hypersensitivity reactions resulting from herbal and conventional toothpaste [11]. Nevertheless, more evidence in this field is needed.

## 3. Oral Antiseptics and Oral Conditions

### 3.1. Dental Caries

Dental caries and periodontal diseases are ranked amongst the most prevalent non-communicable pathologies worldwide [5,112], both being associated with the presence of biofilm [113]. Even though the mechanical control of dental biofilm stands as the most common form of oral hygiene [6,7,8], factors such as limited access, absence of motivation, use of orthodontic appliances and poor oral hygiene abilities can make this control inefficient [13,14,114]. The removal and prevention of the accumulation of microbial plaque may be facilitated by the chemical anti-plaque mouth rinses [7,10,11,12,13,14]. There remains solid evidence corroborating the effectiveness of antiseptic mouth rinse used in addition to mechanical plaque control to decrease or manage plaque and gingivitis [22,115].

Although it has been established that CHX reduces *Streptococcus mutans* (*S. mutans*) colonies [28,93], recent evidence has found that its actual impact on caries prevention remains controversial [116,117]. Therefore, the use of CHX mouthwashes is neither recommended nor restricted [116]. On the other hand, OCT has proven certain value in the prevention of white spot lesions [89,118]. Furthermore, high-quality evidence reports that TRC/copolymer toothpaste leads to a small reduction in coronal caries [64]. However, some worldwide-recognized brands banned toothpaste containing TRC as of early 2019 because of concerns about the possibility of developing endocrine disorders or even cancer [64,119].

Among the EOs, their dominant compounds are monoterpenes, and they have been shown to have a strong antibacterial activity against caries-related microorganisms and, therefore, have emerged as a promising source with potential application in the management of dental caries [79].

### 3.2. Periodontal Diseases

Gingivitis and periodontitis are a continuum of the same inflammatory disease; thus, the efficient treatment of gingivitis stands as the main strategy to effectively prevent periodontitis [120]. When managing gingivitis, the use of anti-plaque agents provided significant improvements in gingival, bleeding and plaque indices [7]. Mouthwashes containing CHX present a highly effective anti-plaque and anti-gingivitis action, followed by EOs and CPC [12,21]. In situations where mechanical plaque control is a challenge, such as post-surgery, necrotizing periodontal diseases or patients in intensive care unit, mouth rinses formulated with EOs and CHX may constitute the right choice [7,121]. In fact, EOs, CHX and TRC may be the best candidates as an adjunct therapy to self-performed mechanical plaque control when the main objective is the control of gingival inflammation [7].

In periodontitis patients, despite the combination of CHX mouthwashes and mechanical debridement, it is possible that subgingival biofilm and calculus remain in the deep pockets [113,122]. A solution to this problem might be a new developing method that consists of the local delivery of antibacterial agents into periodontal pockets [123,124]. However, there are still no significant improvements with this procedure when added to non-surgical periodontal treatment [122].

Another compound that has also shown a marked reduction in salivary bacterial load is OCT [51]. In fact, OCT exhibits a greater reduction in the total count of *Lactobacillus* and *S. mutans* when compared to CHX or PVP-I [118], and it was either superior or comparable to CHX in controlling dental plaque [89]. The use of the 0.1% OCT mouthwash has similar antibacterial efficacy as daily mechanical oral hygiene measures [87,88]. Thus, it is considered an efficient approach for the maintenance of gingival health in the absence of personal mechanical plaque control [51,89]. In the treatment of periodontal disease of patients with HIV, the additional administration of OCT contributed to a significant decline in the bleeding and probing pocket depth values, and atypical microorganisms were eliminated within 1 month post-treatment, even though better results were not reported at the 3-month assessment [89,125]. Therefore, OCT may be a safe and efficient alternative to CHX because of its slightly lower cytotoxic potential [103,126].

Another antiseptic that efficiently improves plaque and gingival inflammatory parameters is CPC, proving a good potential to compensate for the limitations of interproximal plaque control [98,99,100,127]. When mixed with zinc lactate and fluoride, CPC showed substantially higher anti-plaque and anti-gingivitis effects when compared with EOs [98]. However, EOs contribute to greater control of plaque and a decrease in gingival inflammation in the interproximal area when compared with CPC alone [80]. In addition, it has been proposed as an alternative to CHX in patients with orthodontic appliances because it causes less staining [77]. The synergic action of CPC with Hyaluronic Acid in preventing plaque showed that it is as effective as CHX while presenting fewer short-term side effects [128].

Among the antimicrobial agents, the dentifrice formulations that have TRC as the active agent showed the most significant antimicrobial effect, especially when combined with the copolymer of maleic acid and polyvinyl methyl ether (PVM/MA), in order to improve its retention in the oral cavity and its solubility in water, resulting in greater substantivity [53,65]. In children of aggressive periodontitis parents, a TRC toothpaste was demonstrated to be efficient in the management of the periodontal condition through the reduction of bleeding on probing, pocket depth, salivary *Aggregatibacter actinomycetemcomitans* (*A. actinomycetemcomitans*) and inflammatory markers [129]. This compound inhibits the subgingival biofilm formation and the overnight plaque [130,131], but its use has been controversial due to possible associations with low neonatal birth weight [132,133].

### 3.3. Peri-Implant Diseases

Even though the rehabilitation of edentulous spaces with implants is a promissory strategy, peri-implantitis and peri-implant mucositis have increased over time, negatively affecting their survival [134,135,136]. A uniform criteria for the diagnosis and treatment of those pathologies has not been clearly established [137]. The real influence of antiseptics in the development and control of peri-implant diseases remains unclear. The use of 0.03% CHX and 0.05% CPC mouth rinses proved some adjunctive benefits in the treatment of peri-implant mucositis [138]. In addition, it was suggested that 0.2% DEL rinse could prevent peri-implant disease development [139]. This result contrasts with others, where the use of DEL in addition to mechanical debridement, did not show additional improvements [86]. It was also registered that the daily use of a TRC/copolymer toothpaste may represent an important adjunctive at-home approach to inhibit peri-implant mucositis [135].

Recent high-quality evidence indicates that the adjunctive use of antiseptics did not provide significant beneficial effects in the treatment of peri-implant mucositis, although it diminished bleeding on probing after non-surgical treatment of peri-implantitis [122,140]. Thereby, there is a lack of evidence to support the use of additional chemical agents for clinical or microbiological improvement in the treatment of peri-implant disease [141,142].

### 3.4. Candidiasis

*Candida albicans* (*C. albicans*) is a commensal fungal microorganism that colonizes the oral cavity and can grow, causing a fungal infection (candidiasis) when the environment changes [5,143]. Candidiasis is more common in immunocompromised populations and in elderly patients with poor hygiene of their prosthetic dentures, causing denture stomatitis [5,28]. Available therapies for oral candidiasis involve the use of topical or systemic antifungal agents [34]. However, a problem has recently emerged about increasing fungal drug resistance to azoles [5,144]. Hence, there is a demand for alternative compounds [5,145]. Antiseptic mouth rinses have proven to be effective against *C. albicans* [60,146]. In addition, CHX and some EOs (cinnamon, E-cinnamaldehyde, linalool, geranium and melissa) are efficient against *C. albicans* biofilm, which is usually difficult to inhibit efficiently [34]. Nevertheless, host cell cytotoxicity is a consideration when developing these new treatments [147]. Furthermore, alternative natural compounds have shown effectiveness against fungal infections caused by *C. albicans*, such as garlic extract rinse and *Salvadora persica* (Miswak) [108,147].

### 3.5. Herpes Simplex Virus

Viral infections can also affect the oral cavity with multiple presentations and have a relatively short duration [148]. These include HSV, varicella zoster virus, HIV, coxsackie virus and paramyxovirus [28,148]. Infections with herpes simplex are the most common and can usually be managed with supportive therapy [148,149].

Current guidelines recommend the use of antimicrobial mouthwashes, such as CHX or hydrogen peroxide, to reduce secondary infection [148]. CHX destroys DNA and RNA viruses and inactivates several lipophilic viruses (e.g., HSV, HIV, influenza virus, cytomegalovirus) [29,60]. Like other antiseptics, it does not exhibit relevant virucidal activity against small uncoated viruses (enteroviruses, polioviruses, papillomaviruses, SARS-CoV-2 virus) [29,148]. *Salvadora persica* (Miswak) mouth rinse also seems to also have virucidal activity against HSV and fungistatic activity against *Candida albicans* [108].

### 3.6. Halitosis

Halitosis is a common condition mostly caused by microbial activity, namely biofilm formation of the teeth and tongue surfaces of the oral cavity [8,150]. Associated causes of oral malodor are tongue coating, systemic diseases, gingival bleeding, periodontal pocket depth and certain medications [151,152]. Antiseptics play a representative role in the management of halitosis, mainly in reducing the levels of halitosis-related bacteria [150,153]. Although several options have been widely studied, recent evidence reported that TRC toothpaste, mouthwash containing CHX and zinc acetate, and CPC mouthwash are suitable candidates to control it [153]. Also, a mouthwash containing 0.05% CHX, 0.05% CPC and 0.14% zinc lactate seems to significantly diminish the organoleptic scores, as well as levels of volatile sulfur compounds in patients with this condition [151]. In addition, OCT has shown good potential in the prevention of halitosis [89].

## 4. Oral Antiseptics and Systemic Diseases

Oral diseases can eventually result in various systemic diseases [154]. Recent literature has begun to show a strong investment in this correlation, and it has found that the impact that the oral cavity has on the body in general is dizzying [155]. Therefore, it is extremely important to maintain a good oral hygiene status so that oral diseases do not cause or potentiate systemic diseases [156].

The way microorganisms are organized into complex biofilm structures helps to protect resident organisms and allows strict anaerobes to survive even in highly oxygenated areas of the oral cavity, such as surface pockets, tongue, saliva, and oral mucosa [157].

As mentioned above, there are several diseases that may arise in the oral cavity, in which the main etiological factors are microorganisms, such as endodontic and periodontal problems [157]. There have been several studies that show a strong correlation between periodontal diseases and systemic diseases [108,155,156,158,159]. Through the periodontium, there is a large number of microorganisms that can enter the systemic pathway and associate with diseases such as diabetes, neurodegenerative diseases, cardiovascular diseases, etc. [155,156,158,159,160].

In the literature, there are several studies analyzing the correlation between the use of oral antiseptics and systemic diseases, such as bacterial endocarditis [161]. Bacterial endocarditis is classified as an inflammation of the endocardium caused by bacteria that enter the circulation, which have the ability to cause deformations in the heart tissue that depend on the genetic characteristics of the individual [162]. There are studies that show that the bacteria that are present in the oral cavity are the same as those present in bacterial endocarditis, such as the *Streptococcus* group [161,163]. Thus, proper disinfection of the oral cavity can help prevent this pathology [164].

## 5. Oral Antiseptics in the Prevention of VAP

Colonization by microorganisms with pathogenic potential or disruption of the physiological microbiota of the oral cavity may play an important role in the evolution of the disease [165,166]. The influence of the oropharyngeal microflora on the development of nosocomial infections such as VAP has been demonstrated [26,37]. VAP is defined as pneumonia in individuals who have a mechanical device to assist or continuously control breathing via tracheostomy or endotracheal intubation within 48 h preceding the appearance of infection, including the weaning period [167]. It has the greatest negative impact on patient outcomes and hospitalization costs when compared to other nosocomial infections [168]. The most significant mechanism for the onset of VAP is the aspiration of colonized oropharyngeal secretions into the lower respiratory tract [43,169].

Reduction of oral bacterial load prior to treatment of VAP is currently considered one of the most relevant criteria for infection control [47,170,171]. It prevents the accumulation, colonization and micro-aspiration of bacteria with high pathogenic potential, such as *Pseudomonas aeruginosa*, *Acinetobacter baumannii* and *Staphylococcus aureus*, in the mouth and oropharynx [172,173]. To prevent oral complications like VAP and to preserve the integrity of oral tissues, sterility of the lower respiratory tract must be maintained [174].

Tooth brushing, mouth rinsing and professional oral care are significant oral health practices to reduce infection [172,175]. Several studies have reported that the incidence of VAP and other microbial infections decreases when using CHX or other antiseptic preparations and brushing the teeth, from 26% to about 18% [122,176,177].

Solid evidence reported that oral care with CHX or PVP-I effectively reduces the rates of VAP when compared with oral care without these antiseptics [178]. This effect was most prominent for CHX 2%; therefore, CHX 2% is considered the gold standard [178,179].

The use of PVP-I as a valuable substitute for CHX reveals to be an efficient oral care practice aimed at lowering the load of potential pathogens and mitigating the potential risk of infection in both community and hospital environments [56,178,180]. Another viable alternative to CHX for decreasing the microbial flora of patients in the intensive care unit due to its herbal components is an *Echinacea* solution [181].

## 6. Oral Antiseptics in the Prevention of Cross-Infection

The question has been raised in the literature as to whether oral disinfectants have the ability to help combat cross-infection, particularly COVID-19 [182,183,184,185]. The most recent studies found on this topic address the possibility of a correlation between SARS-CoV-2 infection and bacteria in the oral cavity [182,183]. The current use of oral disinfectants will enable a large number of bacteria and viruses responsible for a variety of cross-infectious diseases to be eliminated [182,183,184,185].

COVID-19 is defined as a viral disease that can cause several types of symptoms, such as cough, dyspnea, fever and muscle pain, and also has the ability to lead to other syndromes, such as cytokine storm syndrome, Kawasaki syndrome and toxic shock syndrome [186]. Among all disinfectants used, CHX was the one that showed the best results in the elimination of bacteria and viruses, thus reducing the risk of cross-infection [184].

## 7. Current Concerns about Oral Antiseptics

Commercially, there is a considerable variety of antiseptic mouthwashes that have been shown to be clinically beneficial in the management of oral diseases [23,187,188]. However, a decrease in patient tolerability and acceptability has been noted, as adverse effects such as unpleasant taste and tooth staining have been associated with some of the available products [10,189]. Several mouthwashes contain alcohol, which has been shown to cause mucosal irritation in some patients [10,187] and is also inappropriate for some due to the association between denatured alcohol and oral cancer progression [190].

The long-term use of antiseptics has raised concerns about the possibility of undesirable changes in the composition of the complex oral microflora, spot colonization and the emergence of resilience [191,192]. The modification of the microbiota through a selective method of pathogen elimination, without negatively affecting the normal flora, should be the effective function of the chemical agents in a mouth rinse. Otherwise, this may lead to an outgrowth of pathogenic organisms [49,193].

## 8. Conclusions

This review discusses the most commonly used oral antiseptics and describes their main characteristics. CHX, EOs, POV-I, CPC, TRC and OCT stand out, and BTC, BAC, DEL and herbal products seem to not have a relevant role in clinical practice. However, to support the clinical applicability of oral antiseptics, future studies of robust designs are needed.

## Figures and Tables

**Table 1 jpm-13-01332-t001:** Characteristics of the major oral antiseptics.

Oral Antiseptics	Classification	Most Common Formulations	Mechanism of Action	Spectrum	Adverse Effects
Povidone-Iodine (PVP-I)	Iodophor solution containing a water-soluble complex of iodine and polyvinylpyrrolidone (PVP)	Local topical solution (7.5%, 10%) Spray (5%) PVP iodine solution Fe-150 [55,56]	Inhibits microbial protein synthesis (oxidizing amino acids and nucleic acids) High bactericidal and virucidal activity profile [56,57]	Broad antibacterial spectrum: Gram-positive and Gram-negative; Bacteria spores, fungi, protozoa and several viruses [56,58]	Thyroid dysfunction Allergic dermatitis, after prolonged skin contact and pruritus Metabolic acidosis Acute renal failure Serious adverse effects are not common [55,56,57,58]
Chlorhexidine (CHX)	Cationic surfactant, bisbiguanide	Oral rinses, aerosols and spray formulations (0.12–0.2%) Gels (0.12–1%) Dental varnishes (1%, 10%, 40%) Toothpaste, gels for cleaning teeth and dental flosses [10,28]	Cationic molecule attaches nonspecifically to negatively charged membrane phospholipids of bacteria. It increases the permeability of the cell membrane Impediment of the bacteria membrane’s ability to spontaneously form microdomains Low concentrations (0.02–0.06%): bacteriostatic activity: Affects the change in the osmotic balance of the bacteria cell. This leads to the release of potassium, phosphorus and other low-weight molecules Higher concentrations (>0.12%): bactericidal Cell death by cytolysis. Cytoplasmic coagulation and precipitation [10,59,60]	Wide: more effective against Gram-positive bacteria and weaker against Gram-negative ones. Active against fungi and some lipophilic viruses [10,61,62,63]	Type I and type IV hypersensitivity reactions followed by severe anaphylaxis Taste alteration (hypogeusia) Pain in mouth and tongue Xerostomia Burning sensation Discoloration of tongue Long-term use: swelling of the parotid gland, oral paresthesia; mild desquamation of the oral mucosa and ulceration/erosions; calculus and extrinsic tooth staining [10,28,62]
Triclosan (TRC)	Nonionic phenolic derivative	Toothpaste and mouthrinses 0.3% [64,65]	Inhibition of the enzyme enoyl-acyl reductase (ENR) transporter protein, anti-inflammatory effects: it acts in the inhibition of the cyclooxygenase/lipoxygenase pathways, host-derived inflammatory mediators such as interleukin (IL) 1b, IL-6, tumor necrosis factor, and prostaglandins; damage the bacterial inner membrane [53,66]	Wide, antimicrobial with activity against Gram-positive and Gram-negative bacteria and fungi [65,67]	Taste alterations and mucosal irritations. Long-term use: antibiotic resistance [54]
Benzethonium chloride (BTC) and Benzalkonium chloride (BAC)	Cationic surfactants from quaternary ammonium salts	BTC mouthrinse 0.2% BAC mouthrinse 0.1%, 0.05% [68,69,70]	Though it has not been exactly determined, it is generally accepted that a long, lipophilic alkyl chain penetrates bacterial cell membranes by binding to the cell wall components to produce leakage of the cytoplasmic material, autolysis, and cell death of bacteria [71,72]	Broad: antimicrobial properties against bacteria, fungi and viruses, except for bacterial endospores [73,74]	Detention of human gingival fibroblasts cell cycle Apoptosis [50]
Cetylpyridinium chloride (CPC)	Monocationic quaternary ammonium	Mouthrinses and toothpaste: 0.05–0.10% [75]	Binds to the phosphate groups of lipids in the cell walls of bacteria. It penetrates the cell and causes membrane damage, which leads to leakage of cell components, disruption of bacterial metabolism, inhibition of cell growth and, finally, cell death [52]	Broad antimicrobial spectrum: most effective against gram-positive pathogens and yeast in particular [52,76]	Very limited: tooth staining, ulcers, gingival irritation, burning sensations [76,77,78]
Essential oils—volatile or ethereal oils (EOs) e.g.,: Menthol and eugenol	Complex mixture of odoriferous, volatile organic compounds produced by aromatic plants	External application is the most effective way to use the majority of EOs (e.g., mouthwashes) [79]	Against bacterial pathogens: Denaturation of bacterial proteins, modifying the permeability of the outer membrane of Gram-negative bacteria and the chelation of cations present in the bacterial cytoplasm, rendering the enzymes inactive. The antibacterial activity of essential oils has severe effects as they may seize the growth of the bacteria (bacteriostatic) or kill bacterial cells (bactericidal). Against fungal pathogens: establish a membrane potential across the cell wall and disrupt the assembly of ATP or disintegration of mitochondrial membrane, interfering with the electron transport system (ETS) [79,80,81,82]	Wide spectrum of antibacterial, antifungal, antiviral and insecticidal fungi and yeast; also, potential to inhibit the growth of drug-resistant microbial strains and antioxidant and anti-inflammatory properties [81,82]	Regarded as GRAS (generally regarded as safe) grade chemicals by The U.S. Food and Drug Administration (FDA) Local skin irritation Allergic contact dermatitis Phototoxicity from reaction to sunlight (some oils) [79,82] The majority of the effects are mild, but there have been cases of serious toxic reactions: neurotoxicity, abortions and pregnancy abnormalities, bronchial hyperreactivity, hepatotoxicity, prepubertal gynecomastia, premature thelarche and their endocrine disrupting properties leading to the induction of premature breast growth in young adolescents [15,79,80,81,82,83]
Delmopinol (DEL)	Amino-alcohol	Oral rinse: 0.2% [84,85]	Increases the humidity of tooth surfaces, binds to hard and soft oral tissues as well as to bacterial surfaces, displaces components from the pellicle and interferes with the build-up and cohesion of plaque by reducing glucan synthesis and glucan viscosity. It may also diminish cell-to-cell adhesion [84,85]	Low antimicrobial properties, prevents plaque formation and possesses plaque dissolving properties [86]	Paraesthesia Numbness and altered taste sensations in the oral mucosa Staining [54]
Octenidine (OCT)	Cationic surfactant, Bispyridinamin	Oral rinses: 0.10 (most used), 0.15, and 0.20% [51,87,88]	Bactericidal action by binding to the negatively charged microbial cell membranes and to soft and hard oral surfaces. It disrupts the phospholipid bilayer and destroys the enzyme systems, causing the cell wall, which results in cytoplasmic leakage and cell death [89,90,91]	Gram-positive and Gram-negative organisms, as well as yeasts [89,92]	Tooth and tongue discoloration Dysgeusia [51,87]

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
