# Peer review of "Revisiting Oral Antiseptics, Microorganism Targets and Effectiveness"

_jpm, 2023, doi:10.3390/jpm13091332_

Round 1

Reviewer 1 Report

it would be better if organic and herbal mouth washes are also included

also mention specific percentages related to the most commonly used mouthwashes along with their benefits and adverse effects

generations of mouth washes are missing

minor revision required

Author Response

We are pleased with the opportunity to revise and resubmit our manuscript “Revisiting oral antiseptics, microorganism targets and effectiveness” (Manuscript ID jpm-2561594).

We are grateful for the editor and reviewers’ comments, all have been considered and taken into profound consideration.

Manuscript changes are highlighted in the revised manuscript. Our point-by-point responses to all comments are detailed below. We are happy to consider further revisions and we thank you for your continued interest in our research will better suit this journal

Reviewer: "it would be better if organic and herbal mouth washes are also included. also mention specific percentages related to the most commonly used mouthwashes along with their benefits and adverse effects"

Our Answer:  We explored the hypothesis of including organic and herbal mouthwash products. However, we have verified that these specific products are still in the in vitro stages, which could confuse the readers and the purpose of this review. Yet, when possible we mentioned such products. For example, line 276 mentions an herbal product.

Reviewer: "generations of mouth washes are missing"

Our Answer: We appreciate this suggestion, but it is beyond the scope of this review. However, if this reviewer and editor find it relevant, we are available to add it in the best interest of readers.

Reviewer 2 Report

Comprehensive and well-written review of oral antiseptics. There are a few comments to be addressed.

Line 91. Replace “tabagism” with a more common term such as smoking or heavy smoking. The reference at the end of this sentence, number 93 (Umesh et al.) does not even mention anything to do with smoking or adverse effects due to external factors. Is this an error?

Line 276. Incomplete sentence. Please rewrite.

Line 319-320. Reference number 26 (Gendreau and Loewy) refers to dental stomatitis caused by Candida albicans. This fungal species generally does not cause VAP and there is no mention of VAP in this reference.

A better reference is: Seok-Mo Heo, Elaine M. Haase, Alan J. Lesse, Steven R. Gill, Frank A. Scannapieco. Clinical Infectious Diseases, Volume 47, Issue 12, 15 December 2008, Pages 1562–1570, https://doi.org/10.1086/593193

Author Response

We are pleased with the opportunity to revise and resubmit our manuscript “Revisiting oral antiseptics, microorganism targets and effectiveness” (Manuscript ID jpm-2561594).

We are grateful for the editor and reviewers’ comments, all have been considered and taken into profound consideration.

Manuscript changes are highlighted in the revised manuscript. Our point-by-point responses to all comments are detailed below. We are happy to consider further revisions and we thank you for your continued interest in our research will better suit this journal

Reviewer: "Comprehensive and well-written review of oral antiseptics. There are a few comments to be addressed."

Our answer: We are thankful for this commentary. We have addressed all comments.

Reviewer: "Line 91. Replace “tabagism” with a more common term such as smoking or heavy smoking. The reference at the end of this sentence, number 93 (Umesh et al.) does not even mention anything to do with smoking or adverse effects due to external factors. Is this an error?"

Our answer: We have replaced "tabagism" with smoking, and indeed there was a typo in the reference 93. We have replaced with the correct one: "93.    Stratul SI, Sculean A, Rusu D, Didilescu A, Kasaj A, Jentsch H. Effect of smoking on the results of a chlorhexidine diglu-conate treatment extended up to 3 months after scaling and root planing-a pilot study. Quintessence Int. 2011 Jul-Aug;42(7):555-63. PMID: 21716983."

Reviewer: "Line 276. Incomplete sentence. Please rewrite."

Our answer: We apologize for this typo. We have completed the sentence that now reads as follows: "Salvadora persica (Miswak) mouth rinse also seems to have also virucidal activity against HSV and fungistatic activity against Candida albicans(108)."

Reviewer: "Line 319-320. Reference number 26 (Gendreau and Loewy) refers to dental stomatitis caused by Candida albicans. This fungal species generally does not cause VAP and there is no mention of VAP in this reference.

A better reference is: Seok-Mo Heo, Elaine M. Haase, Alan J. Lesse, Steven R. Gill, Frank A. Scannapieco. Clinical Infectious Diseases, Volume 47, Issue 12, 15 December 2008, Pages 1562–1570, https://doi.org/10.1086/593193"

Our answer: We have replaced accordingly and we appreciate this valid suggestion.

Reviewer 3 Report

The author discussed the commonly used oral antiseptics, describing their characteristics and application in the treatment and prevention of various oral diseases. However, there are several drawbacks which can not be ignored in writing this review:

1. Through the text, the author described the status of researches in a vague way. Supported data and specific description of representative studies are lack.

2. There is only one table in the text, more figures or images are needed.

3. In the section of Conclusions, the author stated that “gives recommendations for their use in the treatment and prevention of various pathologies”. However, little information related to this content can be found in the text.

Requires moderate editing of the English language.

Author Response

We are pleased with the opportunity to revise and resubmit our manuscript “Revisiting oral antiseptics, microorganism targets and effectiveness” (Manuscript ID jpm-2561594).

We are grateful for the editor and reviewers’ comments, all have been considered and taken into profound consideration.

Manuscript changes are highlighted in the revised manuscript. Our point-by-point responses to all comments are detailed below. We are happy to consider further revisions and we thank you for your continued interest in our research will better suit this journal

Reviewer: "The author discussed the commonly used oral antiseptics, describing their characteristics and application in the treatment and prevention of various oral diseases. However, there are several drawbacks which can not be ignored in writing this review:

1. Through the text, the author described the status of researches in a vague way. Supported data and specific description of representative studies are lack."

Our Answer: We respectfully disagree with this statement. We believe that we have clearly reported the current state of research. We were very surprised by this comment because we cited a very wide range of studies (from primary studies to systematic reviews). It would be very useful to have more concrete and specific examples of where these reviewers found such situations so that we can strengthen this manuscript. We look forward to it.

Reviewer: "2. There is only one table in the text, more figures or images are needed."

Our Answer: We found useful to present that table to summarize the information. Regarding the rest, although we are always keen to present figures and images we believe, at this stage, they would not add much to the manuscript. Still, if this reviewer and editor find it relevant we are happy to add further images and figures in the best interest of readers.

Reviewer: "3. In the section of Conclusions, the author stated that “gives recommendations for their use in the treatment and prevention of various pathologies”. However, little information related to this content can be found in the text."

Our Answer: Thank you for pointing this out. We have removed that particular sentence.